# Multimodal Long-Term Predictors of Outcome in Out of Hospital Cardiac Arrest Patients Treated with Targeted Temperature Management at 36 °C

**DOI:** 10.3390/jcm10061331

**Published:** 2021-03-23

**Authors:** Erik Roman-Pognuz, Jonathan Elmer, Frank X. Guyette, Gabriele Poillucci, Umberto Lucangelo, Giorgio Berlot, Paolo Manganotti, Alberto Peratoner, Tommaso Pellis, Fabio Taccone, Clifton Callaway

**Affiliations:** 1Department of Anesthesia and Intensive Care, Azienda Sanitaria Universitaria Giuliano Isontina, University of Trieste, Strada di Fiume 447, 34100 Trieste, Italy; umberto.lucangelo@asuits.sanita.fvg.it (U.L.); berlotg@virgilio.it (G.B.); a.peratoner@gmail.com (A.P.); 2Department of Emergency Medicine, University of Pittsburgh, Pittsburgh, PA 15213, USA; elmerjp@upmc.edu (J.E.); guyefx@upmc.edu (F.X.G.); callawaycw@upmc.edu (C.C.); 3Department of Radiology, Azienda Sanitaria Universitaria Giuliano Isontina, 34128 Trieste, Italy; gabrielepoillucci@gmail.com; 4Department of Neurology, University of Trieste, 34100 Trieste, Italy; paolo.manganotti@asuits.sanita.fvg.it; 5Department of Intensive Care, Azienda Sanitaria Friuli Occidentale Tommaso, 33170 Pordenone, Italy; tommaso.pellis@asfo.sanita.fvg.it; 6Department of Intensive Care, Erasme Hospital, Université Libre de Bruxelles, 1070 Bruxelles, Belgium; fabio.taccone@ulb.ac.be

**Keywords:** cardiac arrest, normothermia, EEG, SSEP, GWR, long term predictors

## Abstract

*Introduction*: Early prediction of long-term outcomes in patients resuscitated after cardiac arrest (CA) is still challenging. Guidelines suggested a multimodal approach combining multiple predictors. We evaluated whether the combination of the electroencephalography (EEG) reactivity, somatosensory evoked potentials (SSEPs) cortical complex and Gray to White matter ratio (GWR) on brain computed tomography (CT) at different temperatures could predict survival and good outcome at hospital discharge and six months after the event. *Methods*: We performed a retrospective cohort study including consecutive adult, non-traumatic patients resuscitated from out-of-hospital CA who remained comatose on admission to our intensive care unit from 2013 to 2017. We acquired SSEPs and EEGs during the treatment at 36 °C and after rewarming at 37 °C, Gray to white matter ratio (GWR) was calculated on the brain computed tomography scan performed within six hours of the hospital admission. We primarily hypothesized that SSEP was associated with favor-able functional outcome at distance and secondarily that SSEP provides independent information from EEG and CT. Outcomes were evaluated using the Cerebral Performance Category (CPC) scale at six months from discharge. *Results*: Of 171 resuscitated patients, 75 were excluded due to missing data or uninterpretable neurophysiological findings. EEG reactivity at 37 °C has been shown the best single predictor of good out-come (AUC 0.803) while N20P25 was the best single predictor for survival at each time point. (AUC 0.775 at discharge and AUC 0.747 at six months follow up). The predictive value of a model including EEG reactivity, average GWR, and SSEP N20P25 amplitude was superior (AUC 0.841 for survival and 0.920 for good out-come) to any combination of two tests or any single test. *Conclusions*: Our study, in which life-sustaining treatments were never suspended, suggests SSEP cortical complex N20P25, after normothermia and off sedation, is a reliable predictor for survival at any time. When SSEP cortical complex N20P25 is added into a model with GWR average and EEG reactivity, the predictivity for good outcome and survival at distance is superior than each single test alone.

## 1. Introduction

After cardiac arrest (CA) most resuscitated patients are comatose as a result of hypoxic ischaemic brain injury [1,2]. Differentiating patients who can awaken from coma from those with irrecoverable injury remains challenging. Bilateral absence of N20 cortical somatosensory evoked potentials (SSEP) at least 72 h after return of spontaneous circulation (ROSC) is one reliable indicator of poor prognosis [3,4,5]. However, the presence of cortical responses on SSEPs does not guarantee favorable outcome [6]. Consensus guidelines recommend that clinicians use SSEP as one part of a multimodal approach to prognostication [4].

There are several gaps in knowledge about SSEP as a prognostic modality after cardiac arrest. First, most studies relate SSEP to short term outcomes like awakening from coma or survival to hospital discharge rather than more important long-term outcomes like function at 3 or 6 months. Second, many studies have used SSEP results in the decision to limit life support for sub-sets of patients, creating bias that would inflate the SSEP performance. Finally, few studies describe how much incremental information SSEP results provide over other clinical information [7,8]. To address some of these gaps Scarpino et colleagues investigated the role of a combination of measures, like SSEP, EEG and GWR, to predict CA patients’s cerebral performance after 6 months follow up [9].

Most studies simplify the complex SSEP waveform into the dichotomous presence or absence of the N20 cortical response. It is not clear whether quantifiable features like latency and amplitude additional also provide prognostic information (Figure 1).

To address some of these gaps, we collected data on the prognostic value of SSEP waveforms performed at regimented times during and after targeted temperature management for predicting outcome after 6 months in a cohort of CA patients with no limitations in life support. We hypothesized that the amplitude of SSEP cortical waveforms is associated with favorable functional recovery, and that SSEP provides independent information from EEG and CT scan. 

## 2. Materials and Methods

### 2.1. Study Design and Population

We performed a retrospective cohort study including consecutive non-traumatic patients resuscitated from out-of-hospital CA who remained comatose (Glasgow Coma Scale score ≤ 8) on admission to an intensive care unit (ICU) at a single center tertiary university hospital from January 2013 to May 2017. Our local ethics committee approved the study and waived the need for informed consent because of minimal risk.

We excluded patients under 18 years old, with traumatic cardiac arrest, with pregnancy, with previously diagnosed progressive neurodegenerative disease, and those deemed likely to meet the criteria for progress to brain death on admission (Figure 2).

### 2.2. Study Definition and Data Collection

We defined cardiac arrest as the abrupt cessation of cardiac activity that required shocks and/or chest compression for the return of spontaneous circulation (ROSC). We dichotomized initial rhythm as shockable (ventricular fibrillation or pulseless ventricular tachycardia) or non-shock-able (pulseless electrical activity or asystole). We estimated the time interval from collapse to ROSC from prehospital reports.

We performed non-contrast enhanced computerized tomography (CT) scan of the brain within 6 h, with a median time of 2 h, from hospital admission to exclude neurological causes of arrest or head trauma in unwitnessed collapse. We treated all patients with targeted temperature management (TTM) to 36 °C for 24 h using an intravenous cooling system (CoolGard 3000/Alsius Icy Heat Exchange Catheter, Zoll Medical, Chelmsford, MA, USA). We sedated patients with propofol, although we substituted mi-dazolam in cases of severe hypotension. We provide neuromuscular blockade with continuous infusion of cisatracurium to prevent shivering during the induction of TTM. We stopped sedation and paralysis at the beginning of the rewarming period.

We measured survival and functional recovery at discharge from the ICU and at 6 months after the event using medical records and cardiologic reports. Reports characterized functional neurological recovery using the Pittsburgh-Glasgow Cerebral Performance Categories scale (CPC), and we defined good outcome as CPC 1–2 [10,11].

#### 2.2.1. EEGs and SSEPs

We acquired SSEPs and EEGs twice, first during the first 12 h after reaching the target temperature of 36 °C (approximately within 12 h after ROSC) and second after withdrawal of sedation and rewarming to 37 °C (approximately 72 h after ROSC).

We recorded EEGs for at least 20 min using a portable machine (Galileo NT PMS version 3.90/00/17014-SPD. EBN Neuro, Firenze, Italy) using a 13 electrodes positioned according to the international 10–20 system [12]. A neurophysiologist determined EEG reactivity as present if there was a clear change in background frequency or amplitude after a pain or voice stimulation [13]. We considered EEGs where electroencephalographic seizures resulted from stimulation to be nonreactive. EEG background was classified as continuous (recognizable clear background activity), discontinuous (burst suppression of almost 10% of the recordings) or flat (isoelectric or sup-pression less than 10 μV) [13]. Epileptiform activity was identified as rhythmic spikes or waves and sharp waves and periodic epileptiform discharges (PEDs) [14,15].

A certified neurologist evaluated peripheral, spinal and cortical SSEPs in response to the stimulation of the median nerve at the wrist. The cortical N20 amplitude was defined as the high-est amplitude of a reproducible potential of CP3/CP4 vs. Fz recordings and CP3/CP4 vs. contralateral earlobe at least at 4.5 ms longer than the previous N13 peak (cervical spinal cord) and within 50 ms after stimulation [16]. We considered as P25 amplitude the first positive wave that follows N20 wave (Figure 1). Consequently we examined the peak to peak N20-P25 absolute amplitude value in both sides. N20-P25 complex has been used previously as a prognostic indicator, particularly in ischemic stroke patients [17,18,19]. In patients with background noise levels under 0.25 μV for whom we could not discern any cortical waveform bilaterally, we considered the SSEPs to be absent. Clinical teams in our hospital do not use SSEP recordings as part of any decisions to withdraw or limit life support.

#### 2.2.2. GWR

We acquired non-contrast head CT scans on Aquilion 64 (Toshiba Medical Systems Europe B.V., Zoetermeer, The Netherlands) or Brilliance iCT 256 (Philips Healthcare, Best, The Netherlands) scanners, using 5 mm slices reconstruction in the axial plane. Two investigators blinded to clinical information examined CT scans for each patient using commercial image viewing soft-ware (suitEstensa Ris Pacs Software, Esaote Healthcare IT, Genova, Italy) with windowing adjusted to “brain,” and identified comparable brain slices at the level of basal ganglia, and at two levels of superior cortex.

Investigators measured average attenuation in Hounsfield Units (HU) of circular regions of measurement (0.1–0.25 cm^2^) using the method described by Torbey et al., with some modification according to more recent reports [20,21]. We recorded HU values bilaterally for gray matter (GM) in the caudate nucleus (CN), putamen (PU), and white matter (WM) in the corpus callosum (CC), and posterior limb of internal capsule (PIC). In particular we chose the anterior halves of posterior internal capsules in order to minimize the density variation of posterior internal capsules between the anterior and posterior halves due to the presence of focal low-attenuation lesions in the posterior half of the posterior internal capsule in 60% of normal brains, according to Choi et al. [22]. We recorded values bilaterally for the medial cortex GM and medial WM at the level of the centrum semiovale (MC1 and MWM1, respectively) and high convexity area (MC2 and MWM2, respectively).

Increasing cerebral edema results in lower attenuation by gray matter and a lower GM to WM ratio (GWR). We calculated GWR basal ganglia = (CN + PU)/(CC + PIC). We calculated GWR cerebrum = (MC1 + MC2)/(MWM1 + MWM2). We calculated average GWR as the mean of the GWR basal ganglia and GWR cerebrum, and we used average GWR for analysis. We divided GWR results into three categories, normal (GWR > 1.2), mild edema (GWR 1.1–1.2), or severe edema (GWR < 1.1), based on prior studies [23].

### 2.3. Statistical Analysis

We describe data with mean (SD) for continuous variables, median (IQR) for non-normally distributed variables, and percentages for categorical variables. We compared variables that differed between patients with good and poor outcome using Chi square or Fisher’s exact test for categorical variables and Wilcoxon rank-sum test for continuous variables.

We tested a correlation of SSEP N20P25 amplitude measurements between the sides and the inter-rater reliability of GWR measurements using Pearson’s correlation. We cross-tabulated GWR, EEG reactivity, and SSEP N20P25 amplitude to examine if any particular finding on one test perfectly predicted the results of another test.

We used logistic regression to test associations between average GWR, EEG reactivity, and SSEP N20P25 amplitude at 36 °C and 37 °C individually and in combination with survival and good outcome at discharge from ICU and after 6 months. Multiple models were created, and the AUC for each compared. With these three predictor variables, we created a total of 13 models: one model with GWR alone and six models at each temperature with SSEP N20P25 amplitude, EEG reactivity, SSEP N20P25 + EEG, SSEP N20P25 + GWR, EEG + GWR or SSEP N20P25 + EEG + GWR.

We used a Hosmer-Lemeshow for goodness of fit for the logistic regression model and a DeLong test to compare the area under the curve (AUC) as each predictor vari-able was added into subsequent logistic regression models. We used a DeLong test comparing AUC both when models included SSEP N20P25 and when the same model was without SSEP N20P25. Statistical analysis were performed using STATA version 15 (Stata Corp, College Station, TX, USA). We considered *p* value ≤ 0.05 statistically significant.

## 3. Results

### 3.1. Baseline Characteristics

Of 171 consecutive patients admitted in our intensive care unit from January 2013 to May 2017, 96 were included in the study. Seventy five patients were excluded due to missing data, uninterpretable neurophysiology data or because patients died during the treatment (Figure 2). Table 1 summarizes baseline and resuscitation characteristics for all subjects and subgroups by ICU survival and outcome. Subjects who died were older. Subjects who died or had poor outcome, received higher total dose of adrenaline and had longer collapse-to-ROSC intervals.

Reliability of SSEPs N20-P25 amplitude measurement between the sides was high with a Pearson’s correlation coefficient of 0.94. Inter rater reliability of GWR was similarly high with Pearson’s correlation coefficient of 0.93.

### 3.2. Association Between GWR, EEG and SSEP

Table 2 and Table 3 describe the association of GWR, EEG and SSEP features. Only one subject (1%) had severe edema on CT scan (GWR < 1.1). This subject had no detectable N20-P25 waveform and no EEG reactivity both at 36 and 37 °C. Among all other subjects, an unfavorable finding in one of these modalities did not perfectly predict unfavorable findings in the others. We measured a strong Pearson correlation coefficient between GWR and SSEP (Pwcorr 0.608). The ROC analysis revealed an AUC for the gray to white average matter ratio (GWRav) at discharge of 0.682 (IC 95% 0.551–0.811) for survival with a very high sensitivity and a very low specificity. The AUC was 0.674 (IC 95% 0.574–0.769) at 6 months with a decrease of sensitivity and an in-crease of specificity. The analysis at 6 months for good outcome demonstrated an AUC of 0.727 (IC 95% 0.629–0.815) Adding SSEP to GWRav to predict survival at discharge, we observed an increase of the AUC up to 0.838 (IC 95% 0.744–0.932) at 37 °C with a slight decrease in sensitivity (91%) and an increase in specificity (56%). The results were similar at 36 °C [AUC 0.781 (IC 95% 0.681–0.880)]. The combinations of SSEP and GWRav at both temperatures were superior to GWR alone. The predictability was also high for good outcome. [AUC 0.835 (IC 95% 0.755–0.914) at 37 °C; AUC 0.823 (IC 95% 0.735–0.908) at 36 °C] (Table 4).

### 3.3. GWR and Outcome

GWRav ranged from 1.07 to 1.45. GWR was higher in subjects who survived relative to those who did not, and in subjects who had good outcome relative to those who had poor out-come (Table 3). The lowest GWRav value among subjects with a good outcome was 1.16, and among subjects who survived was 1.15. The median amplitude for cortical complex SSEP N20P25 with GWR ≥ 1.2 at 36 °C was 0.63 and 0.66 at 37 °C. Median N20P25 amplitude were higher in the mild edema GWR group compared to the no edema group (Table 2).

### 3.4. EEG and Outcome

EEG reactivity to stimulus at 37 °C after rewarming and withdrawal of sedation was present in 41 patients, of whom 31 (69%) survived and 29 (83%) had good outcome at 6 months. There were six patients without EEG reactivity at 37 °C who subsequently survived with a good outcome at 6 months (Table 3).

### 3.5. SSEP and Outcome

Amplitudes of the N20P25 waveform at 36 °C and at 37 °C were larger in subjects who survived relative to subjects who did not survive in each time point with a *p* value < 0.001. Amplitudes were also larger in subjects with good outcome relative to subjects with poor outcome respectively with a *p* value of 0.0001 for N20P25 at 36 °C and a *p* value < 0.001 at 37 °C (Table 3).

### 3.6. Prognostic Value of Combined Modality

Table 4 shows the AUC for the models of EEG reactivity, average GWR, and SSEP N20P25 amplitude alone and in combination for predicting good outcome and survival at discharge at 6 months and the survival at discharge from ICU. EEG reactivity was the best single predictor of good outcome while N20P25 was the best single predictor for survival at each time point. We predicted high probability of survival and good outcome with GWR > 1.15, EEG reactivity present, and SSEP N20P25 amplitude > 0.83 μV.

Predictive value of each combination of two tests was superior to any test alone. Predictive value of a model including EEG reactivity, average GWR, and SSEP N20P25 amplitude was superior (AUC 0.841) for survival and 0.920 for good outcome) to any combination of two tests or any single test.

## 4. Discussion

We demonstrate that when SSEP cortical complex N20P25 is added into a model with GWR average and EEG reactivity, the predictive value for good outcome and survival at distance is superior than when using each single test alone. Furthermore, the predictive value is high also for survival at discharge from ICU. Few prior studies analysed EEG and SSEP recordings in the same patients and, to the extent of our knowledge, none of them added GWR in the model to predict good outcome and survival at distance [14,20,21,22,23].

Few isolated cases of N20 false positive prediction have been reported. Often a retrospective analysis of the data confirmed that was due to background noise levels or misinterpretation [24,25]. Furthermore, the level of tolerance in noise levels remains still unclear [7]. In order to overcome the risk that noise misinterpretation could lead to erroneous conclusions of absent N20, our findings suggest using the total amplitude of the SSEPs wave from the lowest value of N20 to the highest of P25. As in a recent paper, we considered the relationship between cortical amplitudes and outcome overcoming the presence absence dichotomy [26].

SSEPs cortical complex has been previously utilized in other clinical situation for predicting outcome, but no previous studies specifically considered cortical complex N20P25 amplitude as an indicator for good outcome and survival at distance from CA [27,28]. Many papers considered the value of SSEPs only in patients who remained in a comatose state after rewarming, whereas we measured SSEPs in all patients.

We found a difference in amplitude on SSEPs between the first recording during the thermal treatment at 36 °C and the subsequent, without sedation, at 37 °C. Although SSEPs are relatively unaffected by sedation, the difference cannot be merely attributed due to the influence of temperature [16]. Sine EEG and SSEPs were collected in two different moments, the recovery over time could have significant contribution in the neurological evolution process.

EEG reactivity after withdrawal of sedation at 37 °C was the best solo indicator for good outcome at distance. SSEP complex at 37 °C was the best solo predictor for survival at any time. Normally sedation affects more EEG reactivity than SSEP [29]. As shown in Table 4, the three tests combined during hypothermia showed a predictivity for outcomes slightly inferior than after normothermia and off sedation, these results suggest caution using prognostic information in the early stages of treatment.

Few amplitude values compatible with a good outcome and poor outcome for the N20-P25 complex were very close, making our estimate potentially imprecise. Prospective validation is needed to achieve a higher certainty in this threshold. A retrospective study has previously investigated the relationship between N20-P25 complex and outcome after CA, and the results showed the amplitude reduction of N20-P25 complex were associated with poor outcome [19].

Although the requisites for amplitude interpretation were well described previously, these data could not consider the difference between cortical and subcortical potentials. Amplitude could be influenced by the intensity and the number of the stimulus, even if we considered a minimum delay after recording. Adding GWR average to SSEPs complex considerably augmented the predictivity for long term outcome [30]. Patients with a hypoxic ischemic encephalopathy manifest a loss of distinction in gray and white matter. Previous studies focused on the presence of cerebral edema in the very early acute period, GWR values are higher when CT scan is performed within the first 6 h [31,32]. We found only one patient with a GWR value compatible with severe edema (GWR < 1.1). In this patient, the corresponding N20-P25 complex amplitude value was zero and there was no reactivity at each temperature we tested. The majority of patients were gathered into the no edema group (GWR ≥ 1.2). We found a higher median N20P25 amplitude in the mild edema GWR group compared to the no edema group. High or gigantic SSEP are generally indicative of acute changes in neuronal excitability. Giant SSEP were observed in patients with progressive myoclonic epilepsy and diseases with similar clinical features that included post-hypoxic myoclonus [33,34]. Early prognostication could lead to a more rapid treatment and eventually referral to rehabilitation programs. Concerning the model for predictivity at 6 months, many factors could influence the survival and the neurological outcome in a so long amount of time. As suggested in other studies, a more extensive evaluation than just CPC score must be considered for recovery evaluation [35].

There are several limitations to this study. First of all, it is a single center retrospective study with a limited sample size that is focused on OHCA patients treated with TTM at 36 °C, with all the implication that this carries with it. Second, studies on prognostication after CA can suffer from self-fulfilling prophecy. We usually do not consider SSEP in order to withdraw any therapy. Despite this, we can’t exclude that single clinicians could be influenced by neurophysiological results. Third, although imaging was performed within six hours from admittance, we should consider GWR as a picture of a process in evolution that could be different for each patient, even in limited time. Fourth, reactivity was determined as a change in EEG background after a pain or voice stimulation, which are not standardized interventions. Although EEG reactivity is considered a strong predictor of awakening, we need to develop standardized methods for testing and interpretations [36].

## 5. Conclusions

Our study suggests a strong association between SSEPs N20P25 complex and very early GWR measurement and EEG reactivity to predict good outcome at six months and survival at any time. Limitations of the study suggest to further investigate this predictor in a large prospective trial. The predictability of long terms outcomes could be influenced by several factors. Despite this, the early detection of all these indicators seems to have value for clinical practice.

## Figures and Tables

**Figure 1 jcm-10-01331-f001:**
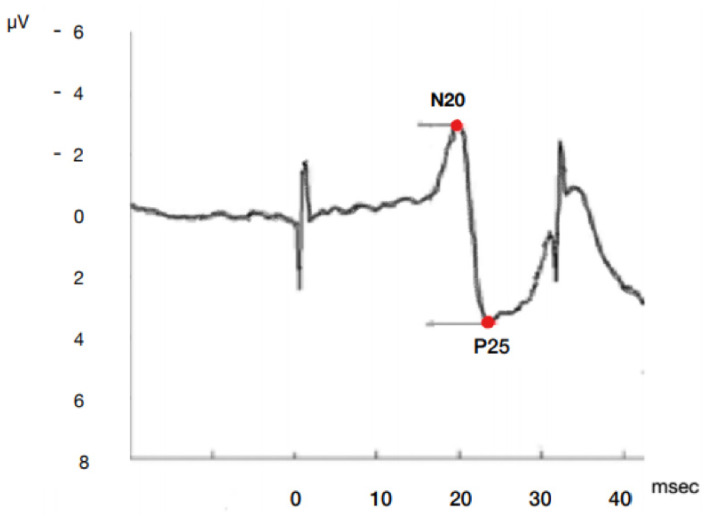
Model of SSEP N20P25 cortical complex wave onset and peak amplitude and latency by median nerve stimulation. µV microvolt, msec milliseconds.

**Figure 2 jcm-10-01331-f002:**
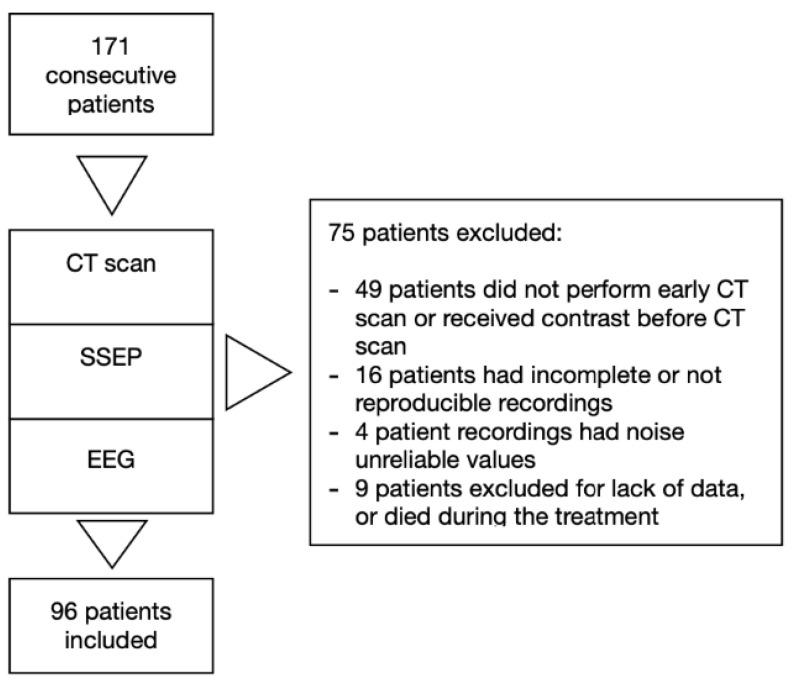
Study design.

**Table 1 jcm-10-01331-t001:** Baseline characteristics and resuscitation details.

	Tot	At Discharge			At 6 Months					
		Survival	Non Survival	*p*	Survival	Non Survival	*p*	Good Outcome	Poor Outcome	*p*
number (*n*, %)	96 (100)	69 (72)	27 (28)		50 (52)	46 (48)		40 (42)	56 (58)	
age (median, IQR)	62 (53–71)	61 (50–67)	72 (60–76)	<0.001	60 (48–66)	67 (60–74)	<0.001	61 (50–67)	72 (60–76)	0.01
**Baseline Characteristics (*n*, %)**										
Male (%)	69 (72)	49 (71)	20 (74)	0.8	37 (74)	32 (70)	0.6	28 (70)	41 (73)	0.7
Hypertension	23 (25)	14 (22)	9 (33)	0.2	10 (21)	13 (29)	0.4	7 (19)	16 (30)	0.2
Coronary artery disease	13 (14)	7 (11)	6 (22)	0.2	5 (11)	8 (18)	0.3	3 (8)	10 (18)	0.2
Chronic heart failure	8 (9)	5 (8)	3 (11)	0.6	4 (8)	4 (9)	0.9	2 (5)	6 (11)	0.3
Diabetes	21 (23)	12 (19)	9 (33)	0.1	9 (19)	12 (27)	0.4	5 (13)	16 (30)	0.07
Chronic kidney disease	7 (8)	5 (8)	2 (7)	0.2	5 (11)	2 (4)	0.3	4 (11)	3 (6)	0.3
Smoker	21 (23)	13 (20)	8 (30)	0.3	10 (21)	11 (25)	0.7	8 (22)	13 (24)	0.8
COPD	9 (10)	5 (8)	4 (15)	0.3	3 (6)	6 (14)	0.2	3 (8)	6 (11)	0.6
**Resuscitation Details**										
Adrenaline mg (mean, IQR)	3 (1–5)	2 (0–4)	4 (2–6)	<0.001	2 (0–3)	4 (2–6)	<0.001	1 (0–3)	4 (2–6)	<0.001
Shockable rhythm (*n*, %)	58 (68)	46 (74)	12 (52)	0.05	34 (74)	24 (54)	0.2	28 (76)	30 (62)	0.2
Time to Rosc minutes (mean, IQR)	17 (12–26)	15 (10–24)	23 (16–29)	0.01	14 (10–20)	23 (16–30)	0.001	14 (10–17)	22 (16–30)	0.0001
*n*° of Shocks during CPR (mean, IQR)	2 (0–5)	2 (1–5)	1 (0–5)	0.1	2 (1–5)	2 (0–5)	0.3	2 (1–4)	2 (0–5)	0.8

**Table 2 jcm-10-01331-t002:** Association between GW ratio and SSEP amplitudes and EEG reactivity at different temperature.

	GWR Average		
	Severe Edema <1.1	Mild Edema 1.1–1.2	No Edema ≥1.2
*n* (%)	1 (1)	17 (18)	78 (81)
EEG Reactivity at 36 °C	0 (0)	4 (27)	23 (31)
EEG Reactivity at 37 °C	0 (0)	4 (25)	37 (51)
N20-P25 amplitude at 36 °C (median, IQR)	0 (0–0)	0 (0–1.13)	0.63 (0–1.73)
If N20-P25 ampl ≠ 0 (median, IQR)	-	1.1 (0.4–1.9)	1.4 (0.7–2.8)
N20-P25 amplitude at 37 °C (median, IQR)	0 (0–0)	0 (0–1.98)	0.66 (0–1.88)
If N20-P25 ampl ≠ 0 (median, IQR)	-	2.3 (0.6–2.7)	1.5 (0.7–2.3)

-: no observation, GWR average: Gray to white matter ratio average.

**Table 3 jcm-10-01331-t003:** Association of GWR, EEG and SSEP features at 36 °C and 37 °C.

*n* (%)	Tot	At Discharge			At 6 Months					
		Survival	Non-Survival	*p*	Survival	Non-Survival	*p*	Good Outcome	Poor Outcome	*p*
**EEG 36 °C (<12 h)**										
Reactivity	27/90 (30)	25 (38)	2 (8)	0.009	20 (42)	7 (16)	0.009	18 (49)	9 (17)	0.001
**Background**										
flat	10 (11)	3 (4)	7 (29)	0.001	2 (4)	8 (19)	0.049	1 (3)	9 (17)	0.049
discontinuous	15 (17)	10 (15)	5 (21)	0.5	5 (11)	10 (23)	0.1	2(5)	13 (24)	0.043
continuous	65 (72)	53 (80)	12 (50)	0.005	40 (85)	25 (58)	0.004	34 (92)	31 (58)	<0.001
**Ictal**										
no discharge	67 (74)	54 (82)	13 (54)	0.007	41 (87)	26 (60)	0.003	35 (95)	32 (60)	<0.001
periodic discharge	11 (12)	5 (8)	6 (25)	0.04	4 (8)	7 (16)	0.38	2(5)	9 (17)	0.1
seizures	12 (13)	7 (11)	5 (21)	0.2	2 (4)	10 (23)	0.008	0 (0)	12 (23)	0.004
**EEG 37 °C (>72 h)**										
Reactivity	41 (46)	39 (61)	2 (8)	<0.001	31 (69)	10 (23)	<0.001	29 (83)	12 (23)	<0.001
**Background**										
flat	12 (13)	3 (5)	9 (36)	<0.001	2 (4)	10 (23)	0.01	0 (0)	12 (23)	0.003
discontinuous	11 (12)	7 (11)	4 (16)	0.5	5 (11)	6 (14)	0.7	3 (9)	8 (15)	0.48
continuous	66 (74)	54 (84)	12 (48)	<0.001	38 (84)	28 (64)	0.027	32 (91)	34 (63)	0.004
**Ictal**										
no discharge	62 (70)	48 (75)	14 (56)	0.09	40 (89)	22 (50)	<0.001	32 (91)	30 (56)	<0.001
periodic discharge	11 (12)	9 (14)	2 (8)	0.46	3 (7)	8 (18)	0.098	2 (6)	9 (17)	0.1
seizure	16 (18)	7 (11)	9 (36)	0.008	2 (4)	14 (32)	1	1 (3)	15 (28)	0.004
**GWR (median, IQR)**										
Basal Ganglia	1.25 (1.2–1.3)	1.27 (1.2–1.3)	1.22 (1.17–1.29)	0.03	1.27 (1.23–1.3)	1.24 (1.2–1.27)	0.002	1.28 (1.2–1.34)	1.23 (1.2–1.27)	<0.001
Average	1.25 (1.2–1.31)	1.26 (1.22–1.3)	1.21 (1.17–1.3)	0.009	1.27 (1.22–1.3)	1–23 (1.2–1.29)	0.004	1.29 (1.2–1.33)	1.23 (1.2–1.27)	<0.001
**SSPEs (median, IQR)**										
n20p25 amplitude at 36 °C	0.48 (0–1.47)	0.83 (0–2.16)	0 (0.0.43)	<0.001	1.17 (0–2.59)	0 (0–0.73)	<0.001	1.41 (0.12–2.8)	0.04 (0–0.94)	0.0001
if n20p25 ampl ≠ 0	1.38 (0.7–2.2)	1.47 (0.8–2.8)	0.6 (0.4–1.4)	0.009	1.9 (0.8–3.5)	0.93 (0.4–1.39)	0.001	2.2 (1.2–3.9)	0.94 (0.4–1.4)	<0.001
n20p25 amplitude at 37 °C	0.52 (0–1.88)	1.06 (0–2.18)	0 (0–0.16)	<0.001	1.49 (0.23–2.3)	0 (0–0.62)	<0.001	1.66 (0.6–2.36)	0 (0–0.68)	<0.001
if n20p25 ampl ≠ 0	1.62 (0.7–2.3)	1.8 (0.9–2.4)	0.48 (0.2–0.7)	0.002	1.93 (1.1–2.6)	0.7 (0.35–1.9)	0.004	1.98 (1.3–2.6)	0.73 (0.5–1.89)	0.003

**Table 4 jcm-10-01331-t004:** AUC for EEG reactivity, average GWR, and SSEP N20P25 amplitude for predicting good outcome and survival.

	Discharge From ICU		At 6 Months			
	Survival		Survival		Good Outcome	
	AUC	(IC 95%)	AUC	(IC 95%)	AUC	(IC 95%)
EEG Reactivity 36 °C	0.647	(0.566–0.729)	0.631	(0.540–0.722)	0.658	(0.562–0.754)
EEG Reactivity 37 °C	0.764	(0.683–0.845)	0.731	(0.626–0.819)	0.803	(0.699–0.875)
GWRav	0.682	(0.551–0.811)	0.674	(0.574–0.769)	0.727	(0.629–0.815)
N20p25 amplitude 36 °C	0.731	(0.629–0.815)	0.707	(0.607–0.797)	0.730	(0.629–0.815)
N20p25 amplitude 37 °C	0.775	(0.674–0.850)	0.747	(0.651–0.833)	0.759	(0.662–0.842)
EEG reactivity 36 °C GWRav	0.783	(0.674–0.893)	0.745	(0.642–0.848)	0.804	(0.709–0.899)
EEG reactivity 37 °C GWRav	0.819	(0.721–0.917)	0.790	(0.697–0.882)	0.872	(0.797–0.947)
EEG reactivity 36 °C N20p25 amplitude 36 °C	0.821	(0.730–0.913)	0.781	(0.687–0.874)	0.831	(0.746–0.917)
EEG reactivity 37 °C N20p25 amplitude 37 °C	0.859	(0.778–0.940)	0.812	(0.722–0.902)	0.887	(0.816–0.958)
GWRav N20p25 amplitude 36 °C	0.781	(0.681–0.880)	0.733	(0.680–0.865)	0.823	(0.735–0.908)
GWRav N20p25 amplitude 37 °C	0.838	(0.744–0.932)	0.798	(0.706–0.890)	0.835	(0.755–0.914)
EEG reactivity 36 °C GWRav N20p25 amplitude 36 °C	0.845	(0.755–0.936)	0.818	(0.731–0.906)	0.882	(0.812–0.953)
EEG reactivity 37 °C GWRav N20p25 amplitude 37 °C	0.882	(0.801–0.963)	0.841	(0.760–0.922)	0.920	(0.864–0.977)

## Data Availability

The data presented in this study are available on request from the corresponding author.

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
