# Peer review of "Multimodal Long-Term Predictors of Outcome in Out of Hospital Cardiac Arrest Patients Treated with Targeted Temperature Management at 36 °C"

_jcm, 2021, doi:10.3390/jcm10061331_

Round 1

Reviewer 1 Report

The present study dealt with important issue on predicting longterm outcome with multimodal approaches. The methods are fair and the conclusions of the study could affect decision process of out-of-hospital cardiac arrest in clinical fields immediately.

However, one issue should be fixed. The authors described that the outcomes were evaluated at six months from discharge in the abstract. If this was true, there were a possibility that the outcomes of the patients could be compared at different time points because hospital stays of the patients might be different with each other. The outcomes of the patients should be compared at the same time points. Usually, the outcomes of out-of-hospital cardiac arrest are compared at 3 or 6 months since return of spontaneous circulation. The authors described the time points of measuring outcomes as "6 months after the event" in the methods section. Therefore, I cannot determine the exact time point of measuring outcomes in this study.

If the authors measured the outcome at six months from discharge, the time point of measuring the outcome should be changed at six  months from return of spontaneous circulation. As a result, the outcome should be remeasured.

If the event in the methods section (6 months after the event) means cardiac arrest day, the measuring time of outcome in the abstract should be corrected with the same time point in the method section.

Author Response

Response to Reviewer 1 Comments

Point 1:However, one issue should be fixed. The authors described that the outcomes were evaluated at six months from discharge in the abstract. If this was true, there were a possibility that the outcomes of the patients could be compared at different time points because hospital stays of the patients might be different with each other. The outcomes of the patients should be compared at the same time points. Usually, the outcomes of out-of-hospital cardiac arrest are compared at 3 or 6 months since return of spontaneous circulation. The authors described the time points of measuring outcomes as "6 months after the event" in the methods section. Therefore, I cannot determine the exact time point of measuring outcomes in this study.

If the authors measured the outcome at six months from discharge, the time point of measuring the outcome should be changed at six  months from return of spontaneous circulation. As a result, the outcome should be remeasured.

If the event in the methods section (6 months after the event) means cardiac arrest day, the measuring time of outcome in the abstract should be corrected with the same time point in the method section.

Response 1: Correct. Outcome has been evaluated six months after the cardiac arrest day. I have added the correct sentence in the abstract. 

Reviewer 2 Report

Although this is a single-center retrospective study, the authors report important information on the early prediction of long-term outcomes in patients resuscitated after cardiac arrest. The combination of EEG reactivity, SSEPs cortical complex N20P25, and Gray to White matter ratio on brain CT can provide highly accurate prognostic information. 
The reviewer has two concerns in the methods section.
Regarding the CT imaging, the images were taken within 6 hours of the patient's arrival. What was the mean (or median) value?
Concerning outcome assessment, were the evaluators blinded to the prognostic test results?
These issues may be study limitations.

Author Response

Response to Reviewer 2 Comments

Point 1:Regarding the CT imaging, the images were taken within 6 hours of the patient's arrival. What was the mean (or median) value?

Response 1: Correct. CT imaging has been performed early after the admission at the emergency department, once the patient has been stabilized. We considered early the CT scan performed until the 6 hours from the arrival at the emergency department. The median value was 2 hours and 17 minutes from the admittance to the hospital.
This limitation has been added in the discussion section. 

Point 2:Concerning outcome assessment, were the evaluators blinded to the prognostic test results?

Response 2: Outcome evaluators were neurologist and intensivist residents not involved in the data collection and in the treatment of the patients.

 These issues may be study limitations.

Response 3: I have added this limitation in the discussion section.